# Intensification of Double Kinetic Resolution of Chiral Amines and Alcohols via Chemoselective Formation of a Carbonate–Enzyme Intermediate

**DOI:** 10.3390/molecules27144346

**Published:** 2022-07-06

**Authors:** Jan Samsonowicz-Górski, Anna Brodzka, Ryszard Ostaszewski, Dominik Koszelewski

**Affiliations:** Institute of Organic Chemistry, Polish Academy of Sciences, Kasprzaka 44/52, 01-224 Warsaw, Poland; jan.samsonowicz-gorski.stud@pw.edu.pl (J.S.-G.); awoltanska@icho.edu.pl (A.B.); ryszard.ostaszewski@icho.edu.pl (R.O.)

**Keywords:** enantioselectivity, lipase, substrate engineering, chemoselectivity, biocatalysis, carbonates

## Abstract

Chiral amines and alcohols are synthons of numerous pharmaceutically-relevant compounds. The previously developed enzymatic kinetic resolution approaches utilize a chiral racemic molecule and achiral acyl donor (or acyl acceptor). Thus, only one enantiodivergent step of the catalytic cycle is engaged, which does not fully exploit the enzyme’s abilities. The first carbonate-mediated example of simultaneous double chemoselective kinetic resolution of chiral amines and alcohols is described. Herein, we established a biocatalytic approach towards four optically-pure compounds (>99% *ee*, Enantioselectivity: *E* > 200) via double enzymatic kinetic resolution, engaging chiral organic carbonates as acyl donors. High enantioselectivity was ensured by extraordinary chemoselectivity in lipase-catalyzed formation of unsymmetrical organic carbonates and engaged in a process applicable for the synthesis of enantiopure organic precursors of valuable compounds. This study focused not only on preparative synthesis, but additionally the catalytic mechanism was discussed and the clear impact of this rarely observed carbonate-derived acyl enzyme was shown. The presented protocol is characterized by atom efficiency, acyl donor sustainability, easy acyl group removal, mild reaction conditions, and biocatalyst recyclability, which significantly decreases the cost of the reported process.

## 1. Introduction

In the foundations of pharmaceutically-relevant molecules often lays a chiral, small-molecular scaffold, which is required to be in enantiopure form. Every stereoisomer of a drug candidate must be studied separately, for the biological activity and potential pharmaceutical application of opposite enantiomers may vary significantly. For instance, (*R*)-1-phenylethylamine is used for the synthesis of Etomidate (an intravenous anesthetic agent), [1] while a (*S*)-1-phenylethylamine substructure can be found in the (*S*)-Rivastigmine utilized in treatment of Alzheimer’s disease [2]. Thus, it is crucial to have access to both enantiomers of chiral compounds. Nevertheless, the predominant method of accessing enantiomerically pure compounds is the kinetic resolution (KR) of racemates promoted by enantioselective enzymes such as lipases, which are able to catalyze acyl transfer reactions [3,4,5,6,7]. To increase reaction yields, DKR (dynamic kinetic resolution) was developed; however, the product configuration is limited by the selectivity of the enzyme. Furthermore, the utilization of a metal-containing racemization catalyst may cause problems with final product contaminations. The classical KR utilized in industry and laboratory synthesis engages only half of the enzyme catalytic potential. Therefore, the full possible use of an enzyme is not obtained. The reactions catalyzed by lipases engage enantiomer differentiation in the first part of the catalytic cycle (KR by hydrolysis) [8] or the second (alcoholysis or aminolysis) [9]. Thus, reactions were enantio-differentiation takes place twice in the catalytic cycle are promising in enantioselective synthesis by KR. Such an approach can double the turnover number (TON) of the process, and increase the reaction yield and atom economy (Figure 1).

Every catalytic cycle consists of acyl-enzyme intermediate formation, followed by reaction with a nucleophile. The combination of these two steps in one process will result in the double use of the enzyme, increasing the catalytic efficiency and atom economy. Nowadays, emphasis is put on the development of protocols giving access to a set of synthetically-relevant products in one-pot processes with great atom efficiency. Therefore, in the course of our studies on the intensification of enantioselective reactions [10,11], we decided to study if it was possible to combine two enantiodivergent steps in one process and to optimize it for preparative purposes.

An approach that fulfils the mentioned requirements is double enzymatic kinetic resolution (DEKR), enabling the simultaneous resolution of two racemates in one-pot, by enantioselective acyl transfer from a chiral acyl donor to a chiral acyl acceptor. A previously published method utilizes double dynamic kinetic resolution (DDKR) [12] and is limited by the selectivity of the enzymes, which may hamper access to the opposite enantiomer of the acyl acceptor. Additionally, the negative impact of the acyl donor in previous protocols was neglected. Reaction of methoxyacetates and propionates results in the formation of esters and amides, whose hydrolysis requires harsh conditions. Alternatively, carbonates are sustainable acyl donors containing two groups, with the possibility of being exchanged. Therefore, one molecule of carbonate can act twice as an acyl donor, increasing the atom economy. Additionally, hydrolysis of carbonates or carbamates takes place under mild conditions [13,14]. The published protocols used vinyl carbonates, which led to the release of toxic acetaldehyde [13,15,16,17,18], and the observed enantioselectivity was moderate in most cases. Moreover, the catalytic cycle and the role of the acyl-enzyme intermediate structure were neglected. Due to the advantages of carbonates as acyl donors, we decided to examine their performance with various methods, with potential synthetic utilization for the stereoselective synthesis of carbonates, carbamates, alcohols, and amines.

## 2. Results and Discussion

### 2.1. Chemo- and Enantioselectivity Studies on Carbonate EKR

Initially, the screening of sixteen enzymes (including hydrolases used in numerous biocatalytic procedures) was performed, finding *Candida antarctica* lipase B (CALB, Novozym 435) as the only biocatalyst with the ability to catalyze the substitution of the alkoxy group of carbonate. In order to determine if the alkoxy groups of dimethyl carbonate (DMC) or diethyl carbonate (DEC) can be substituted with alcohol enantioselectively, alkoxy-group transfer reactions were performed with a high excess of alcohol on model substrates: DMC as acyl donor and 1-phenylethanol (**1a**) (Appendix A). Asymmetric carbonate was obtained exclusively, despite a ratio of 1:10 DMC to alcohol, indicating that only one alkoxy group of the carbonate can be exchanged in a lipase-catalyzed process. Thus, we concluded that the carbonate-derived acyl-enzyme intermediate was formed chemoselectively. The analysis of model reaction products showed that the enzyme is not only chemoselective towards mixed carbonate, but also highly enantioselective. Among the previously published protocols regarding enzymatic synthesis of mixed carbonates, enantioselectivity was neither analyzed nor optimized [19]. To fill this gap, the optimization of reaction conditions was performed (Appendix A, Section: Enzymatic kinetic resolution of alcohols). MTBE (methyl *tert*-butyl ether) was found to be the most suitable solvent (providing higher conversions than those obtained under neat conditions), and room temperature (22–25 °C) was optimal, with a molar 1:5 ratio of alcohol to carbonate. Under the optimized conditions, (*R*)-methyl 1-phenylethyl carbonate (29% yield, *ee* > 99%) and (*R*)-methyl 1-phenylethyl carbonate (14% yield, *ee* > 99%) were obtained. A higher conversion rate was obtained when DMC was used as acyl donor (Appendix A); therefore, it was chosen for the scope and limitation studies (Table 1).

**Table 1 molecules-27-04346-t001:** Enzymatic enantioselective acylation of alcohols with DMC ^1^.

Entry	*rac*-1 ^2^	Conv. (%)	(*S*)-1 *ee* (%)	(*R*)-2 *ee* (%)	*E* ^3^
1	**1a**	31	40	>99	>200
2	**1b**	3	2	>99	>200
3	**1c**	13	14	>99	>200
4	**1d**	60	6	4	1
5	**1e**	14	16	>99	>200
6	**1f**	37	52	>99	>200
7	**1g**	44	45	55	5
8	**1h**	48	84	>99	>200
9	**1i**	35	67	>99	>200
10	**1j**	45	81	>99	>200
11	**1k**	23	31	>99	>200
12	**1l**	55	8	7	1
13	**1m**	87	*rac* (+/−)	*rac* (+/−)	*nd*
14	**1n**	93	*nd*	*nd*	*nd*

^1^ Reaction conditions: **1a–n** (0.26 mmol), DMC (1.3 mmol, 5 equiv.), and MTBE (1 mL), r.t. 24 h., shaker 200 rpm, ^2^ According to the description in Figure 2 and Figure 1. ^3^ Enantioselectivity calculated on equation: *E* = ln[(1 − *ee*_s_)/(1 + *ee*_s_/*ee*_p_)]/ln[(1 + *ee*_s_)/(1 + *ee*_s_/*ee*_p_)].

**Figure 1 molecules-27-04346-f001:**
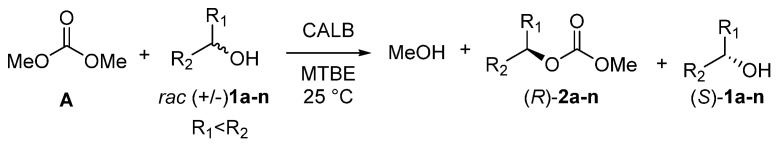
Structures of studied compounds: **1a-o**: R=H; **2o-a**: R=C(O)OCH_3_; **3a-o**: R=C(O)OC_2_H_5_.

Subsequent scope and limitation studies using the selected secondary and primary alcohols (Figure 1 and Table 1) showed that (*R*)-carbonate esters of secondary alcohols can be obtained with high enantioselectivity (*E* > 200), while the hydroxyl group was placed between the small group (such as methyl) and medium or large group (*n*-butyl or phenyl). EKR of primary alcohols with a hydroxyl group on the alpha position resulted in the synthesis of carbonate with low enantioselectivity (*E* = 1), which can be explained by the greater distance between the catalytic center and stereogenic center. Notably, β-citronellyl methyl carbonate was obtained with a 87% yield; however, the product was racemic. Therefore, when the stereogenic center is placed on position 3 (beta position), the reaction is not selective, due to the greater distance from the catalytic triad. The observed synthesis of asymmetric carbonates can be explained by the high chemoselectivity exhibited by the used enzyme [20,21]. An analogous reaction, performed with a chemical catalyst, resulted in the mixture of symmetrical disubstituted and asymmetric carbonates. Racemization studies indicated that Novozym 435 is enantiospecific, since secondary (*S*)-alcohols were not accepted by the enzyme and carbonate products were optically pure (>99% *ee*), despite a long incubation with the enzyme. Thus, the developed method provides access to enantiopure 2-hexanol (an after product of carbonate hydrolysis), which is hard to obtain using the previously published protocols [22]. In a situation where the enzyme is not enantiospecific but enantioselective, racemization can be observed when equilibrium reactions (such as reversible acyl transfer) are used in biocatalysis (Appendix A, Section: Enantiospecificity studies).

The observed results of alcoholysis with racemic alcohols acknowledge the utilization of enzymes as efficient catalysts for the synthesis of enantiopure unsymmetrical carbonates (and alcohols, after carbonate group removal). Organic carbonates are employed for synthesis of pharmaceuticals [23], ester oils [24], or the intermediates of fine chemicals [25,26]. The successful optimization of enzyme-catalyzed carbonate synthesis enables replacing reactions between chloroformate (unstable and harmful) and the desired alcohol, which require carcinogenic pyridine [27,28] or energy consuming chemical catalytic processes, suffering from a lack of enantioselectivity [26,29,30,31,32]. The presented protocol functions in one enantiodivergent step per catalytic cycle.

### 2.2. Carbonate-Mediated DEKR of Alcohols

The observed chemo- and enantioselectivity encouraged us to examine the alcoholysis of asymmetric racemic carbonates with racemic chiral alcohols. Due to the fact that the selected enzyme accepts only (*R*)-enantiomers of substrates, alcoholysis of racemic carbonate is prone to providing (*S*)-carbonate and (*R*)-alcohol. As a result, double enzymatic kinetic resolution of alcohols and carbonates was performed, with methyl 1-phenylethyl carbonate (**2a**) and 2-hexanol (**1h**) as model substrates. Solvent screening showed MTBE to be the solvent providing the highest conversion; however, its enantioselectivity was poor. Toluene at 50 °C was found to be optimal for the DEKR procedure, and Novozym 435 was found to be the only enzyme able to catalyze the process efficiently (Appendix A). Model carbonates methyl 1-phenylethyl and ethyl 1-phenylethyl were used as acyl donors and compared. For most alcohols the reaction was enantioselective, and depending on the alcohol-acyl acceptor, the enantioselectivity of acyl removal from the carbonate substrate differed. The results are presented in Table 2.

Differences in the enantioselectivity of the first step proved that the catalytic cycle steps depend on one another and that the acyl acceptor impacts on the enantioselectivity of acyl group removal **2/3a**. Notably, when **2a** was used as acyl donor, the enantioselectivity of first step of DEKR was lower than the second. When **3a** was used, the situation was reversed (2-hexanol was an exception). Thus, we presume that the formation of an enzyme-methyl carbonate intermediate is responsible for the enantioselectivity enhancement or reduction in the second step. The obtained results indicated that the intensification of the kinetic resolution efficiency was successful for most alcohols, and the studied reactions were enantioselective. The formation of mixed carbonate with two chiral alkoxy groups was not observed. To obtain a deeper insight into the catalytic cycle of the studied process, we decided to examine the alcoholysis of model carbonates with ethanol and methanol in 1-molar equivalents (Table 2, Entry 14, 15). An exchange of a less sterically-hindered group was observed, which indicates the presence of different acyl intermediates in an equilibrium driven process. A productive process depends not only on the enzyme selectivity, but also on the nucleophile structure. The small alkoxy group (methoxy or ethoxy) of carbonate is only exchanged by another small alcohol (i.e., methyl is exclusively exchanged with ethyl). More sterically-hindered alkoxy groups (such as 1-phenylethoxy) are substituted by high molecular weight alcohols (such as 2-hexanol or 1-phenylpropanol) enantioselectively. However, no enantiospecific process of alcoholysis was observed, and every reaction led to racemization products when the reaction time was longer than the optimal 96 h.

### 2.3. Carbonate-Mediated DEKR of Alcohols and Amines

To determine if DEKR with chiral carbonates can be used for the resolution of amines, we decided to study the scope of aminolysis of carbonates. According to the published literature concerning the utilization of carbonates in stereoselective procedures, symmetric carbonates (DMC and DEC) [18] and mixed vinyl carbonates were utilized as acyl donors in enantioselective synthesis of carbamate with a chiral alkoxy group [14,15]. The products were obtained by satisfying *ee*; however, the reaction lacked enantiospecificity, and toxic acetaldehyde was formed as a byproduct.

By combining enantioselective acyl transfer from model carbonates with acylation of chiral amine, a double enzymatic kinetic resolution was established. Model studies of the established process were performed on 1-phenylethyl amine (**1o**) and methyl 1-phenylethyl carbonate (**2a**). The optimization of reaction conditions revealed toluene at 50 °C to be most suitable and Novozym 435 (CALB) as the only efficient enzyme catalyzing reaction studied (Appendix A). To our great delight, (*R*)-methyl (1-phenylethyl) carbamate was obtained enantiospecifically, with a 45% yield of isolated product under optimized conditions. Moreover, (*R*)-1-phenylethanol was obtained with 90% *ee* and (*S*)-methyl 1-phenylethyl carbonate with 89% *ee*.

The possibility of reusing the enzyme notably reduces the cost of the process. Therefore, the recyclability of Novozym 435 was examined. The enzyme recyclability in the fifth run gave 56% of the initial activity, without loss of enantioselectivity (Figure 2 and Appendix A). Notably, during runs 1–3, the activity decreased only slightly; therefore, the catalyst can be used three times for this process. Further recycling resulted in a decrease of reaction yield. However, the high enantioselectivity (whose decrease was not observed during recycling) indicates the possibility of the utilization of enzyme recycled more than two times. With the optimal conditions in hand and the catalyst characterized, we performed further enantioselectivity studies to obtain a deeper insight into the DEKR process. Under optimal conditions, a dependence of the *ee* values on the conversion was recorded for the model reaction (Figure 3).

The model reaction stopped near a conversion of 50%, which shows enantio-specificity towards (*R*)-1-phenylethylamine. Moreover, product formation was not observed when (*S*)-methyl 1-phenylethyl carbonate was used as a substrate. Analogously, when (*S*)-1-phenylethylamine was used as substrate, its conversion was not observed, despite a long incubation time. Having in hand general and efficient conditions for the one-pot synthesis of four compounds with high enantioselectivity, we examined the potential for optimized reaction on various alcohol derivatives and two model amines (Table 3).

To investigate the impact of a small molecular group of asymmetric carbonates on enzyme enantioselectivity (*E*), ethyl and methyl carbonates of 1-phenylethanol (**2/3a**) and 1-phenylpropanol (**2/3b**) were examined (Table 3, Entry 1–4). The obtained results indicated that methyl serving as smaller molecularly-unchangeable group is preferred by the enzyme, enhancing not only the enantioselectivity but also the conversion, similarly to DEKR by carbonate alcoholysis. In the case of a reaction between 1-phenylethylamine (**1o**) and a **2/3a-b** derivatives ethyl group, negative impacts on reaction selectivity and velocity were found (reaction between **1p** and **2b** was three-fold slower than the model reaction). Notably, (*R*)-1-phenylprop-2-en-1-ol (**1c**) was obtained with a higher yield. Since a double bond is known to be shorter than a single, **2c** fits better in the catalytic center of the enzyme. Thus, the developed method enabled enantioselectively obtaining (*R*)-1-phenylprop-2-en-1-ol, whose structural motif can be found in Reboxetine (an important antidepressant sold as drug Edronax^®^). Further studies on the scope and limitations showed that high enantioselectivities (*E* > 200) toward (*R*)-alcohols were obtained for secondary alcohols with a hydroxyl group near the chirality center. However, enantioselectivity towards aminolysis of (*S*)-carbonate was observed when the hydroxyl group was placed on the alpha of the stereogenic center. Therefore, when methyl 2-phenylprop-1-yl carbonate (**2d**) was used, (*S*)-alcohol was obtained (Table 3, Entry 6). The presented approach enabled synthesizing precursors of biologically active substances, such as novel glucokinase activators (**(*R*)-1k**; (*R*)-1-phenylpropan-2-ol) [33], potassium channel blockers (**(*R*)-1j**; (*R*)-4-phenylbutan-2-ol) [34]; or chiral cyclopropane derivatives’ precursor (**(*R*)-1i**; (*R*)-4-phenylbut-3-en-2-ol) [35]. All these compounds were obtained with a high enantiomeric excess and in a form easy to separate and use for further syntheses, after carbonate group removal under mild conditions. The easiness and efficiency of the separation of the product states allowed for high recovery yields, with a range 92–98% for all products (for a conversion of 50%) (Appendix A). To investigate the mentioned impact of carbonate chirality on the reaction enantioselectivity, *n*-hexyl methyl carbonate (**2n**) was used as an acyl donor. The reaction remained enantiospecific toward (*R*)-1-phenylethylamine; thus, we presume that the enantio-specificity of amine acylation comes from the carbonate asymmetry and carbonate–enzyme intermediate structure. To explore the enantioselectivity of developed method, the scope of DEKR of carbonates was evaluated with 4-phenylbutan-2-amine (precursor of the compounds used for treatment of hemangiomas or vascular malformations) [36] as acyl acceptor. Racemization of the products (**2a** and **3a** in case of EKR, **2o** and **2p** in case of DEKR) was not observed, and the *ee* remained high, despite the long reaction time (see Appendix A).

After scope limitation studies and a thorough analysis of the reaction outcomes, the impact of carbonate as an acyl donor on the reaction course was discussed. The proposed four-step catalytic cycle shows two chemoselective and two enantioselective steps (Figure 3). The results of our studies indicated that the steps responsible for the binding of the acyl donor in the catalytic pocket (**I**) and alcohol release (**II**) have an enantioselectivity lower than the second step where the acyl group is transferred onto the amine (**III**). Thus, we presume that the binding of the C(=O)OCH_3_ group in the catalytic center is responsible for this enantioselectivity enhancement. Moreover, the enantioselectivity of C(=O)OCH_3_ moiety removal was enhanced by carbonate asymmetry, as shown by the results obtained for 1-hexyl methyl carbonate (**2n**). Thus, a clear impact of enzyme–carbonate acyl intermediate (**C**) on the reaction selectivity was demonstrated. During the first step, when the intermediate **C** is formed, its structure depends on the structure of the compound responsible for the nucleophilic attack in the third step. This was proven by the alcoholysis of the model carbonates with ethanol and methanol. In earlier works on carbonate double enantioselective reactions, an acyl–enzyme intermediate was formed selectively due to the irreversible tautomerization of vinyl alcohol. Thus, the more sterically-hindered intermediate was productive instead of the methyl/ethyl, which was not formed due to the utilization of vinyl-derived substrates. Thus, we proved that two types of intermediate can be formed in equilibrium driven processes. In the studied processes, only one form was productive, in contrast to chemically-catalyzed processes. The productivity of the intermediate depends not only on the enzyme selectivity, but also on the structure of nucleophile acting as an acyl acceptor in the second enantio-divergent reaction (**III**,**IV**).

Due to the enzyme selectivity and substrate nucleophilicity, the used biocatalyst forces the attack of the more sterically-hindered nucleophile when a low molecular weight intermediate is formed. Enzyme chemoselectivity results in an increase of enantioselectivity (sometimes resulting in enantio-specificity). Therefore, we showed that with careful substrate engineering, the enantioselectivity can be efficiently increased and two kinetic resolutions can be performed in a one-pot process. The presented protocol shows a clear correlation between chemoselectivity and enantioselectivity, which has been not discussed previously concerning the catalytic cycle. Previously, the utilization of vinyl carbonates forced the reaction equilibrium towards more sterically-hindered products, which resulted in a lower stereoselectivity. Our method is an example of substrate engineering leading to a notable increase of enantioselectivity through engagement of chemoselectivity.

### 2.4. Upscale Studies

Following the scope and limitation studies, the upscaling of the selected reactions was examined. Studies performed at 10 mmol scale clearly indicate that the developed protocols of EKR and double enzymatic kinetic resolution can be utilized for synthetic purposes (Appendix A). The former provides a useful access to enantiopure (*R*)-carbonates (i.e., **(*R*)-2h** 47% conversion, 44% yield, >99% *ee*) and *S*-alcohols, the latter results in obtaining various compounds with high enantioselectivity. Notably, after upscaling, the enantioselectivities of the studied reactions remained as high as they were at analytical scale. In the case of the aminolysis of compounds **2f,h–j**, the reactions were highly enantioselective towards both *R*-amine and *R*-carbonate after upscaling to 10 mmol. Notably, alcohol **(*R*)-1h** was synthesized with a high yield and >99% *ee,* which is very difficult to obtain using acyl donors such as vinyl acetate [22]. The compound **1h** was recognized as a synthon of substances such as δ-lactones [37], inhibitors of soluble epoxide hydrolase [38], and FICA (novel derivatives used for determining the absolute configuration of secondary alcohols using ^19^F and ^1^H NMR spectroscopy) [39].

## 3. Materials and Methods

### 3.1. Materials and Instrumentation

All the chemicals were obtained from commercial sources, and the solvents and GC gases were of analytical grade. ^1^H and ^13^C NMR spectra were recorded in CDCl_3_ solution using a Bruker Oxford 400 NMR spectrometer (400 MHz) (Oxford Instruments, Bristol, England). Chemical shifts are expressed in parts per million using TMS as an internal standard. MS spectra were recorded on an API 365 mass spectrometer (SCIEX) (A B Sciex, Alderley, England). Conversion studies were performed on a Gas Chromatograph PerkinElmer Clarus 680 instrument (PerkinElmer Ltd., Beaconsfield, England), which was also used for enantiomeric excess determination; nitrogen as carrier gas, with temperature gradient variants provided in Section 3.6, split 40:1, flow 1 mL/min, column: CP 7502 CP-Chiralsil DexCB 25 m, 0.25 mm, 0.25 μm. Enantiomeric excesses were determined using a HPLC system by SCION instruments: Series 6000 CO-6310 (SCION Instruments, Goes, The Netherlands), with a diode array detector DA-6430 (chromatograms recorded at λ_1_ = 220 nm) at temperature 20 °C (or 15 °C as mentioned) on Chiralcel OD-H column or Reprosil Chiral-OM column; with hexane: isopropanol serving as eluent (flow: 1 mL/min) with a PU-6100 pump. Determination of conversion when benzylamine was used as an acyl acceptor was performed using a Gas Chromatograph PerkinElmer Clarus 680 (PerkinElmer Ltd., Beaconsfield, England), using a column VF1701ms (Agilent Technologies, Cheadle, England), 30 m length, 0.25 mm cross-section, equipped with FID detector, nitrogen as carrier gas, flow 1 mL/min, split 40:1, with a temperature gradient 70–140 °C (rate 10 °C/min), hold of 140 °C for 3 min, 140–200 °C (rate 10 °C/min), and hold of 200 °C for 2 min at the end of the run (method 3). Structural and chromatographic data are attached in the Appendix A. Optical rotations were measured on a Jasco P-2000 polarimeter at 25 °C, λ = 589 nm, with chloroform as solvent and cuvette 39.35/PX/50, Borofloat (Dunmow, England), path length 50 Starna scientific Ltd. For incubation a Thermo scientific MaxQ4450 shaker (Warsaw, Poland), was used. Enzymes used for screening: Novozym 435 CALB (*Candida antarctica* lipase B, activity: 10,000 PLU/g) were purchased from Novo Nordisk. PPL (crude lipase type II), wheat germ lipase, Amano PS, Amano AK, Acylase from *Aspergillus meleus*, Lipozyme, crude Lipase from *Candida antarctica*, lipase from *Rhizopus arrizus*, and lipase from *Candida cylindracea* were purchased from Sigma-Aldrich (Merck, Darmstadt, Germany). Immobilized enzymes: Tl imino (*Thermomyces lanuginosa* lipase), PS imino (*Pseudomonas cepacia* lipase), CalA imino (*Candida antarctica* lipase A), CR imino (*Candida rugosa* lipase), and RM imino (*Rhizomucor miehei* lipase) were purchased from Purolite^®^ Life Sciences (Llantrisant, England). The native enzymes set: Chiral Enzyme Spectrum “Amano” Ver. 2 was obtained from Amano.

### 3.2. Carbonate Synthesis

A total of 10 mmols of the corresponding alcohol was put in a round bottom flask and dissolved in 30 mL of DCM. Then, 11 mmol of pyridine was added and the flask was stirred in dry ice–acetone bath for 10 min. Subsequently, 17 mmol of chloroformate was added dropwise. After completion of chloroformate addition, the reaction mixture was stirred at room temperature for 4 h. The reaction mixture was washed with 30 mL 1 M HCl. Then water phase was extracted with DCM (3 × 15 mL). The combined organic phases were dried over anhydrous MgSO_4_. Then, the solvent was removed on a rotary evaporator, and the residue was purified by column chromatography (hexane/ethyl acetate), to obtain pure carbonate [27]. Solvents were carefully removed on a rotary evaporator at room temperature, as carbonates are likely to evaporate in vacuo.

### 3.3. Enzymatic Kinetic Resolution of Alcohols

A total of 3 mL vial alcohol (0.26 mmol) and dimethyl carbonate (1.25 mmol) were dissolved in 1 mL of MTBE, the enzyme was added (50 mg), and the reaction was placed on a shaker (200 rpm, rt.). Conversions were calculated using GC measurements (sample preparation: 50 µL of reaction mixture dissolved in 1 mL of ethyl acetate).

### 3.4. Double Enzymatic Kinetic Resolution of Alcohols

In a 3-mL glass vial of carbonate (0.26 mmol) 1-phenyletylamine (0.26 mmol) was dissolved 1 mL of toluene, and Novozym 435^®^ CALB (50 mg) was added. The vial was closed and put on shaker (200 rpm, 50 °C) for three days. Then, 50 µL of reaction mixture was dissolved in 1 mL of ethyl acetate for GC studies. The remaining mixture was purified by column chromatography (hexane: ethyl acetate) (Appendix A). For all reactions, two E-values for the two steps of kinetic resolution were calculated using the general calculation method [22]. According to the limitations of method accuracy, E-values above 200 were stated as “>200”. The optical purity of amines was calculated with optical rotation or upon derivation with an acetyl group.

### 3.5. Upscale Procedures

In a glass vial were placed Novozym 435^®^ CALB (100 mg), carbonate (2 mmol), and 1-phenylethylamine (2 mmol), and 3 mL of toluene was added. The vial was closed and put on a shaker (200 rpm, 50 °C) for three days. Then, 50 µL of reaction mixture was dissolved in 1 mL of ethyl acetate for the GC studies. The remaining mixture was purified by column chromatography (hexane:ethyl acetate). Similarly, when the model reaction was performed on the model substrates at a scale of 10 mmol of carbonate and 10 mmol of amine, the enantioselectivity remained the same as obtained in the scope studies.

### 3.6. Enantioselectivity Determination

The enantioselectivities of the reactions were determined using GC-FID with a column CP7502 CP-Chirasil Dex CB 25 m 0.25 mm, 0.25 µm (Agilent Technologies), with a temperature gradient from 60 °C to 200 °C, and with a rate of 5 °C/min, with a hold of 200 °C for 3 min at the end of the run (method 1) or rate 3 °C/min with hold of 200 °C for 3 min at the end of the run (method 2) (general method profile is presented in Materials and Instrumentation section). In the case of compounds that could not be separated on the abovementioned column, chiral HPLC or optical rotation was used (columns OD-H or OM). Optical rotation measurements was also used to determine the *R*/*S* configuration of the alcohols obtained in the double enzymatic kinetic resolution reaction in cases where the standard HPLC or GC was used or where chromatographic separation was unsuccessful.

## 4. Conclusions

In summary, highly efficient protocols for double enzymatic kinetic resolution, leading to four enantiopure compounds (>99% *ee*), were successfully established. The aim of engaging chemo- and stereoselectivity in the development of a sustainable method for the synthesis of four enantiopure compounds in a one-pot process was successfully achieved. The developed method, by engaging complementary reactions, enables the versatile and enantioselective synthesis of amines and alcohols in a one-pot process. Additionally, the reaction time and amount of reagents required for the efficient separation of enantiomers were significantly decreased. In the course of our studies, we showed that only one alkoxy group of methyl- or ethyl carbonate derivatives is exchanged chemoselectively. The obtained results indicate that careful substrate engineering is crucial for the course and mechanism of the biocatalytic process. By substitution of vinyl carbonates with methyl- or ethyl carbonates, not only was toxic acetaldehyde release eliminated, but additionally the enantioselectivity was dramatically increased, giving access to all enantiomers of secondary amines and secondary alcohols with a hydroxyl group on the chiral center [13,14,15]. The presented protocol demonstrates that the utilization of chiral racemic mixed carbonates as acyl donors was responsible for the notable enantioselectivity improvement. The synthetic potential of the established protocol was shown on a group of alcohols and amines. The *E*-values of acyl transfer are higher for a kinetic resolution of acyl acceptor, which suggests that the second step of the DEKR process is enhanced by a low-molecular weight carbonate group being bound in the catalytic center. By combining the chemoselectivity towards a more sterically-hindered group of mixed carbonate with the enantioselectivity of enzyme-catalyzed acyl transfer, the enantioselectivity of the whole process was efficiently increased. Moreover, the enzyme–carbonate intermediate was responsible for the enantioselectivity increase in the acyl-removal part of the catalytic cycle. Thus, the enantioselectivity of acylation by carbonates and the second part of DEKR were improved. Moreover, the atypical tendency for exchanging an carbonate alkoxy group more structurally similar to the nucleophile was observed. An explanation was found by analysis of catalytic cycle studies and the chemical equilibrium. Our studies and previously published papers suggested that asymmetric carbonate forms two acyl–enzyme intermediates (by exchanging one of two possible alcoxy groups). The presented studies proved that through nucleophile choice one of these two intermediates can be productive. Thus, with selection of substrates, the composition of the final reaction mixture can be modified.

The presented approach combines two complementary reactions in a one-pot process, thus providing easy access to all enantiomers of pharmaceutically relevant products, with great atom economy. The deprotection of the obtained carbamates and carbonates under mild conditions eliminates possible racemization and increases the sustainability of the proposed process. The catalyst can be recycled and reused up to five times, without loss of enantioselectivity. The successful upscaling and utilization of environmentally sustainable carbonates provides the possibility of implementing the developed protocol in industrial applications.

## Data Availability

On request by those interested.

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
