# Peer review of "Intensification of Double Kinetic Resolution of Chiral Amines and Alcohols via Chemoselective Formation of a Carbonate–Enzyme Intermediate"

_molecules, 2022, doi:10.3390/molecules27144346_

Round 1

Reviewer 1 Report

“Intensification of double kinetic resolution of chiral amines and alcohols via chemoselective formation of carbonate-enzyme intermediate” is an extensive and interesting work providing an efficient and innovative methodology to access enantiopure alcohols and amines. In the introduction, authors well described the meaningful and the aim of their study. However, results and discussion part as well as S.I need to be reorganized to help reader to better follow the methodology employed. That is why a recommend major revisions.

Part 2.1 opens on the description of reaction with an excess of alcohol, no procedure was reported in S.I., despite the mention of (Supporting information). In fact, I finish by finding it in S.I page 6. Results in ESI concerning this part has to be given in the same order than results were presented in the main text, starting with the experiments with excess of alcohol. Precise in the text the number of screened enzymes, will be appreciable to not have to go and back to material and methods section. Give the yield for DMC and DEC of your optimized procedure with MTBE. It is necessary to introduce the scope of your limitations study, at least, with the direct and indirect enantioselectivity expected with secondary alcohol vs primary alcohol on alpha or beta position of a stereogenic center. In table 1, footnotes include DEC however, no results were presented with this acyl donor, this mention has to be removed. Same as previously, you mentioned many times results with DEC but I find them only on page 7 of S.I. At least, if you can’t reorganize all S.I, indicate where the corresponding results can be found (e.g. Supporting information Table S3).

L.101 result with β-citronellol have to be explain in regard of the position of the stereogenic center

L.106 I think it is necessary to precise that (S) selectivity is for secondary alcohol. For compounds with primary alcohol, the selectivity could be different.

The last sentence in S.I. about enantiospecificity should be included in the manuscript and refer to S.I for details.

In general, supporting information have to be more factual without conclusions on results, these are expected in the main text. Tables have to be numbered S1, S2, S3 etc instead of S1a, S1b, S2. Another general remark, when using enzymes, it is better to work regarding the specific activity of each enzyme. Indeed, 50 mg of one can lead to significantly different activity compare to 50 mg of another.

Part 2.2

As previously the selectivity for (S) is for secondary alcohol. Refer to the right section in S.I. when you talk about optimal conditions (Toluene, 50 °C) which are different from the ones of the previous section. Figure S2 didn’t bring more information than table S4, remove it. As previously, the conclusion on solvent screening are in SI, this is not its place. In one sentence, it is necessary to mention that you have optimized also enzymatic amount and evaluated the recyclability referring to the right part of S.I. In this section, no reference was made to S.I whereas 5 pages where dedicated to this one in S.I. For alcoholysis reaction, did methanol or ethanol was in excess? It have to be precise in S.I. In addition, discussion have to be rephrased. What high molecular weight alcohol? Example will be a plus to enhance comprehension.

Part 2.3

L178, 1-phenylethyl amine is 1o and not 1n.

Have you optimized conditions for amine or used same as alcohol? Include data for amine in S.I if different from these already reported from alcohol. In addition, it is not clear if recyclability of the enzyme was performed for DEKR of alcohol or amine-alcohol or both. Refer in the text to the right S.I figures.

L211, precise (R)-1-phenylprop-2-en-1-ol.

L217, precise that alcohol is on the carbon on alpha of the chiral center.

L220-221 (R)-1K, (R)-1J and (R)-1I.

It will be interesting to give isolated recovery yield of each product for at least one example to assess the “easy to separate” claims L223.

L235 remove the “,” between your subject and the verb

L245-246. Sentence need to be rephrased.

Scheme 3, please conserved same orientation of the molecule, in order to not have bond front or back depending of where is the molecule for a single enantiomer.

Even if it is a small part, upscaling has to be distinguished from the part 2.3 as you upscale both EKR and DEKR. As previously, S.I. has to be updated to follow the general organization of the main manuscript.

L273 (R)-1H

Part 3.

I was surprised by the use of nitrogen in GC instead of helium or hydrogen often used, could you confirm this point? In addition, give the gradient for the raise of temperature.

Material and methods have to be carefully reformatted according to change in S.I to be consistent and no redundant.

Conclusion

L.373 all enantiomers of secondary alcohols and amines. Limitation studies proved that is not thecase for alcohol which are not directly on the chiral center.

S.I NMR description lack of attribution of peaks

Author Response

Dear Editor,

please find enclosed for your consideration the revised article molecules-1785567 entitled  “Intensification of double kinetic resolution of chiral amines and alcohols via chemoselective formation of carbonate-enzyme intermediate” by Koszelewski et al.

               Firstly, we would like to express our gratitude to Reviewers for their suggestions that allowed us to considerably improve our manuscript. We have revised the text according to the suggestions and we hope that you will now find it suitable for publication in the Molecules. Below, please find the detailed information on the changes in the manuscript with answers to all comments. All changes made in the manuscript were highlighted in yellow.

Reviewer 1: Intensification of double kinetic resolution of chiral amines and alcohols via chemoselective formation of carbonate-enzyme intermediate” is an extensive and interesting work providing an efficient and innovative methodology to access enantiopure alcohols and amines. In the introduction, authors well described the meaningful and the aim of their study. However, results and discussion part as well as S.I need to be reorganized to help reader to better follow the methodology employed. That is why a recommend major revisions.

Response: We are very grateful for this remark. Due to the Reviewer suggestions results and discussion part as well as S.I parts were carefully revised and modified.

Reviewer 1: Part 2.1 opens on the description of reaction with an excess of alcohol, no procedure was reported in S.I., despite the mention of (Supporting information). In fact, I finish by finding it in S.I page 6. Results in ESI concerning this part has to be given in the same order than results were presented in the main text, starting with the experiments with excess of alcohol.

Response: Thank you very much for these remarks. Order of procedures concerning optimization of reaction conditions and studies on selectivity in S.I. is matched to the order of main manuscript. All numbers of tables were provided in references to S.I. We are very grateful for noticing the “DEC” in Table 1 footnote which was removed. Examples and experimental data stating for advantage of DMC over DEC are presented in S. I. In scope and limitation of carbonate EKR the differences in enantioselectivity expected (and obtained) between primary and secondary alcohol were additionally described. Results with DEC are presented in S.I. DEC in main article mentioned once in order to show that it was studied and was removed where it was not suitable to mention about it.

Reviewer 1: Precise in the text the number of screened enzymes, will be appreciable to not have to go and back to material and methods section. Give the yield for DMC and DEC of your optimized procedure with MTBE. It is necessary to introduce the scope of your limitations study, at least, with the direct and indirect enantioselectivity expected with secondary alcohol vs primary alcohol on alpha or beta position of a stereogenic center.

Response: We are very grateful for this remark. Due to the Reviewer suggestions the number of screened enzymes was provided. The yield for carbonates of optimized reaction was provided. The scope of primary and secondary alcohols together with the appropriate comment to this was provided in the main manuscript.

Reviewer 1: In table 1, footnotes include DEC however, no results were presented with this acyl donor, this mention has to be removed. Same as previously, you mentioned many times results with DEC but I find them only on page 7 of S.I. At least, if you can’t reorganize all S.I, indicate where the corresponding results can be found (e.g. Supporting information Table S3).

Response: We are very grateful for this remark. Due to the Reviewer suggestions discussion regarding results with DEC was revised and modified. Information regarding results with DEC were clarified and the place where they are placed in S.I was provided.

Reviewer 1: L.101 result with β-citronellol have to be explain in regard of the position of the stereogenic center

Response: Thank you very much for this remark. The explanation of lack of enantioselectevity of citronellyl methyl carbonate synthesis in regard of the position of the stereogenic center was provided.

Reviewer 1: L.106 I think it is necessary to precise that (S) selectivity is for secondary alcohol. For compounds with primary alcohol, the selectivity could be different.

Response: Thank you very much for this remark. Novozym 435 is (R)-selective enzyme. In each case, for this reason in each case we obtained products with (R)-configuration and (S)-enantiomer remained unreacted. This selectivity remains in agreement with literature data regarding immobilized lipase from Candida antarctica (e.g. Synthesis 2009(10):1725-1731, DOI:10.1055/s-0028-1088121; J. Mater. Sci. 2018, 53(3), DOI:10.1007/s10853-018-2641-5). The enantiopreference of this enzyme is underlined in the text.

Reviewer 1: The last sentence in S.I. about enantiospecificity should be included in the manuscript and refer to S.I for details.

Response: Thank you for your suggestion. The sentence was included in the manuscript and reference to S.I. was provided.

Reviewer 1: In general, supporting information have to be more factual without conclusions on results, these are expected in the main text. Tables have to be numbered S1, S2, S3 etc instead of S1a, S1b, S2. Another general remark, when using enzymes, it is better to work regarding the specific activity of each enzyme. Indeed, 50 mg of one can lead to significantly different activity compare to 50 mg of another.

Response: Thank you very much for this remark. S.I. was correct to be more factual and conclusions were transferred to the manuscript or remove when unnecessary. Tables are numbered S1, S2, S3 etc… Thank you for the suggestion about specific activity. Although, in studied process only Novozym 435 catalyzes reaction (for other screened enzymes activity in studied processes was not observed). The specific activity of this enzyme is standardized during process of production. Therefore the same amount of immobilized enzyme exposes the same activity. The specific activity of Novozym was provided in “Material and Instrumentation” section.

Reviewer 1: As previously the selectivity for (S) is for secondary alcohol. Refer to the right section in S.I. when you talk about optimal conditions (Toluene, 50 °C) which are different from the ones of the previous section. Figure S2 didn’t bring more information than table S4, remove it. As previously, the conclusion on solvent screening are in SI, this is not its place. In one sentence, it is necessary to mention that you have optimized also enzymatic amount and evaluated the recyclability referring to the right part of S.I. In this section, no reference was made to S.I whereas 5 pages where dedicated to this one in S.I. For alcoholysis reaction, did methanol or ethanol was in excess? It have to be precise in S.I. In addition, discussion have to be rephrased. What high molecular weight alcohol? Example will be a plus to enhance comprehension.

Response: Thank you very much for this remark. We want to apologize for not providing the optimization procedure for DEKR by carbonate alcoholysis. Now all experimental data concerning optimization is provided and references to S.I. was provided in the manuscript. Figure with activity with different solvents and conclusions about solvent screening with were removed. Conclusion is provided in the manuscript. The recyclability of enzyme was not studied for DEKR by alcoholysis. Alcohols and carbonates were always used at 1:1 ratio. Examples of alkoxy group (“high molecular weight” and “low molecular weight” alcohol-derived) are provided to enhance comprehension.

Part 2.3

Reviewer 1: L178, 1-phenylethyl amine is 1o and not 1n.

Response: Thank you very much for this remark. It was corrected.

Reviewer 1: Have you optimized conditions for amine or used same as alcohol? Include data for amine in S.I if different from these already reported from alcohol. In addition, it is not clear if recyclability of the enzyme was performed for DEKR of alcohol or amine-alcohol or both. Refer in the text to the right S.I figures.

Response: Thank you very much for this remark. The section regarding mentioned studies was provided in manuscript together with modified S.I.

Reviewer 1: L211, precise (R)-1-phenylprop-2-en-1-ol.

Response: Thank you very much for this remark. It was corrected.

Reviewer 1: L217, precise that alcohol is on the carbon on alpha of the chiral center.

Response: Thank you very much for this remark. It was corrected.

Reviewer 1: L220-221 (R)-1K, (R)-1J and (R)-1I.

Response: Thank you very much for this remark. It was corrected.

Reviewer 1: It will be interesting to give isolated recovery yield of each product for at least one example to assess the “easy to separate” claims L223.

Response: Thank you very much for this suggestion. Isolated recovery yields for model reaction products were mentioned and shortly discussed in the manuscript. Table with experimental data concerning recovery studies is provided in S.I.

Reviewer 1: L235 remove the “,” between your subject and the verb

Response: Thank you very much for this remark. It was corrected.

Reviewer 1: L245-246. Sentence need to be rephrased.

Response: Thank you very much for this remark. It was corrected.

Reviewer 1: Scheme 3, please conserved same orientation of the molecule, in order to not have bond front or back depending of where is the molecule for a single enantiomer.

Response: Thank you very much for this remark. It was revised and modified due to the Reviewer suggestion.

Reviewer 1: Even if it is a small part, upscaling has to be distinguished from the part 2.3 as you upscale both EKR and DEKR. As previously, S.I. has to be updated to follow the general organization of the main manuscript.

Response: Thank you very much for this remark. Upscaling was distinguished for part 2.3 and in S.I. was organized in one section.

Reviewer 1: L273 (R)-1H

Response: Thank you very much for this remark. It was corrected.

Part 3.

Reviewer 1: I was surprised by the use of nitrogen in GC instead of helium or hydrogen often used, could you confirm this point? In addition, give the gradient for the raise of temperature.

Response: We confirm that we used nitrogen as carrier gas. Nitrogen was efficient as a carrier gas for our studies, sufficient separation was reached, moreover we could decrease the cost of analysis. Temperature gradients for methods were provided in Section 3.6 and other instrumentation parameters gathered in Materials and Instrumentation section.

Reviewer 1: Material and methods have to be carefully reformatted according to change in S.I to be consistent and no redundant.

Response: Thank you very much for this remark. It was revised and corrected due to the Reviewer suggestions.

Conclusion

Reviewer 1: L.373 all enantiomers of secondary alcohols and amines. Limitation studies proved that is not the case for alcohol which are not directly on the chiral center.

Response: Thank you very much for this remark. Due to the Reviewer suggestion this part was revised and rephrased.

Reviewer 1: S.I NMR description lack of attribution of peaks

Response: Thank you very much for this remark. The description of NMR spectra was prepared in accordance with the requirements set by the journal and was prepared analogously to other published works (Molecules 2022, 27(11), 3633; https://doi.org/10.3390/molecules27113633; Molecules 2018, 23(9), 2241; https://doi.org/10.3390/molecules23092241).

Reviewer 2 Report

The authors report the double kinetic resolution of chiral amines and alcohols via chemoselective formation of carbonate-enzyme intermediate. The presented concept is very interesting and, in my opinion, the current manuscript deserves to be published after addressing the next comments:

1) I recommend to include the part with recycling of the enzyme into the main text and discuss it.

2) The Table 2 is very cumbersome, and it is very difficult to understand it.

2) Before publication, English should be improved significantly.

Author Response

Dear Editor,

please find enclosed for your consideration the revised article molecules-1785567 entitled  “Intensification of double kinetic resolution of chiral amines and alcohols via chemoselective formation of carbonate-enzyme intermediate” by Koszelewski et al.

               Firstly, we would like to express our gratitude to Reviewers for their suggestions that allowed us to considerably improve our manuscript. We have revised the text according to the suggestions and we hope that you will now find it suitable for publication in the Molecules. Below, please find the detailed information on the changes in the manuscript with answers to all comments. All changes made in the manuscript were highlighted in yellow.

Reviewer 2  The authors report the double kinetic resolution of chiral amines and alcohols via chemoselective formation of carbonate-enzyme intermediate. The presented concept is very interesting and, in my opinion, the current manuscript deserves to be published after addressing the next comments:

Reviewer 2: 1) I recommend to include the part with recycling of the enzyme into the main text and discuss it.

Response: Thank you very much for this suggestion. The part with enzyme recycling was provided in the manuscript.

Reviewer 2: 2) The Table 2 is very cumbersome, and it is very difficult to understand it.

Response: Thank you very much for this remark. We agree with the Reviewer's opinion that the table is very complex. In our opinion, this is the best form of comparing the results for easier comparative analysis. Such presentation of the results obtained from numerous experiments gives the potential reader an insight into the scope and effectiveness of the presented method of synthesizing non-racemic compounds.

Reviewer 2: 3) Before publication, English should be improved significantly.

Response: Thank you for this remark. As suggested by the Reviewer, the work (main article and S.I.) were carefully checked in terms of language. We made every effort to correct the expressions and phrases used, as well as various language errors.

Round 2

Reviewer 1 Report

Thank you for this new improved version. Just some little points to clarify:

L101. As ee>99% were obtained for product with both DMC and DEC, I suppose that the higher enantioselectivity is for the starting material and this is a direct consequence to a better conversion. In my opinion, sentence has to be rephrased or the term enantioselectivity has to be removed to avoid misunderstanding.

Scheme 3. There is a error of configuration when molecule was bind with the enzyme in B and D. Check carefully.

In S.I please give the list of enzymes instead of refering to the main text. In addition check carefully if tables and figures are correctly numbered in the text. I find page 19, Table S9 in the text whereas in fact it is Table S11.

L106-107. Please add a sentence for the primary alcohol in alpha of a stereogenic center.

Thanks again for this interesting work

Author Response

Firstly, we would like to express our gratitude to Reviewers for their suggestions that allowed us to considerably improve our manuscript. We have revised the text according to the suggestions and we hope that you will now find it suitable for publication in the Molecules. Below, please find the detailed information on the changes in the manuscript with answers to all comments. All changes made in the manuscript were marked up using the “Track Changes” function.

Reviewer 1: Thank you for this new improved version. Just some little points to clarify: Thanks again for this interesting work

Response: Thank you very much for the effort Reviewer put into reviewing our work. Thanks to these tips, we significantly improved the quality of the manuscript. Thank you for expressing a positive assessment of our work.

Reviewer 1: L101. As ee>99% were obtained for product with both DMC and DEC, I suppose that the higher enantioselectivity is for the starting material and this is a direct consequence to a better conversion. In my opinion, sentence has to be rephrased or the term enantioselectivity has to be removed to avoid misunderstanding.

Response: Thank you for this remark. The term enantioselectivity was removed because we agree that in context of this phrase and data it refers it could cause misunderstanding.

Reviewer 1: Scheme 3. There is a error of configuration when molecule was bind with the enzyme in B and D. Check carefully.

Response: We are very grateful for this remark. Indeed, there were mistakes in configuration of compounds S1, P1 and P2. Errors were corrected and checked to ensure that correct enantiomers were depicted.

Reviewer 1: In S.I please give the list of enzymes instead of refering to the main text. In addition check carefully if tables and figures are correctly numbered in the text. I find page 19, Table S9 in the text whereas in fact it is Table S11.

Response: Thank you very much for this remark. List of enzymes was provided in S.I. and table numbers and references were corrected and checked.

Reviewer 1: L106-107. Please add a sentence for the primary alcohol in alpha of a stereogenic center.

Response:  Thank you very much for this suggestion. There was a mistake in sentence below describing selectivity of primary alcohols and “beta” was incorrect thus it was changed into “alpha” to give proper sense of the sentence. Therefore, there is clearly outlined that stereogenic center was placed in alpha carbon atom.

Reviewer 2 Report

The authors have revised the manuscript taking into consideration the reviewers’ comments and the article can be published in Molecules.

Author Response

Firstly, we would like to express our gratitude to Reviewers for their suggestions that allowed us to considerably improve our manuscript. We have revised the text according to the suggestions and we hope that you will now find it suitable for publication in the Molecules. Below, please find the detailed information on the changes in the manuscript with answers to all comments. All changes made in the manuscript were marked up using the “Track Changes” function.

Reviewer 2: The authors have revised the manuscript taking into consideration the reviewers’ comments and the article can be published in Molecules.

Response: Thank you very much for the effort Reviewer put into reviewing our work. Thanks to these tips, we significantly improved the quality of the manuscript.
